# Enhanced Antitumoral Activity of Encapsulated BET Inhibitors When Combined with PARP Inhibitors for the Treatment of Triple-Negative Breast and Ovarian Cancers

**DOI:** 10.3390/cancers14184474

**Published:** 2022-09-15

**Authors:** Alberto Juan, María del Mar Noblejas-López, Iván Bravo, María Arenas-Moreira, Cristina Blasco-Navarro, Pilar Clemente-Casares, Agustín Lara-Sánchez, Atanasio Pandiella, Carlos Alonso-Moreno, Alberto Ocaña

**Affiliations:** 1Centro Regional de Investigaciones Biomédicas (CRIB), University of Castilla-La Mancha, 02008 Albacete, Spain; 2Translational Oncology Laboratory, Translational Research Unit, Albacete University Hospital, 02008 Albacete, Spain; 3Facultad de Farmacia de Albacete, Universidad de Castilla-La Mancha, 02008 Albacete, Spain; 4Facultad de Ciencias y Tecnologías Químicas, Universidad de Castilla-La Mancha, 13005 Ciudad Real, Spain; 5Centro de Investigación del Cáncer—CSIC, Instituto de Investigación Biomédica de Salamanca, CIBERONC, 37007 Salamanca, Spain; 6Experimental Therapeutics Unit, Hospital Clínico San Carlos, Fundación Para la Investigación Biomedica de El Hospital Clínico San Carlos, CIBERONC, 28040 Madrid, Spain

**Keywords:** olaparib, JQ1, combination regimes, ovarian cancer, breast cancer, nanomedicine, polymeric nanoparticles, liposomes

## Abstract

**Simple Summary:**

Poly (adenosine diphosphate ribose) polymerase inhibitors (PARPis) have demonstrated antitumoral activity in several cancers harbouring germline and somatic BRCA1/2 mutations. The widespread use of these agents in clinical practice is restricted by the development of acquired resistance due to the presence of compensatory pathways. A strategy to deal with this is the use of combination therapies with drugs that act synergistically against the tumour. BETis can completely disrupt the HR pathway by repressing the expression of BRCA1 and could be aimed at generation combination regimes to overcome PARPi resistance and enhance PARPi efficacy. However, this strategy is hampered by the poor pharmacokinetic profile and short half-life of BETis. In this work and as a proof of concept, we discuss the potential preclinical benefit provided by the combination of the PARPi olaparib and the BET inhibitor JQ1 encapsulated into nanoparticles for the treatment of BRCAness tumours.

**Abstract:**

BRCA1/2 protein-deficient or mutated cancers comprise a group of aggressive malignancies. Although PARPis have shown considerably efficacy in their treatment, the widespread use of these agents in clinical practice is restricted by various factors, including the development of acquired resistance due to the presence of compensatory pathways. BETis can completely disrupt the HR pathway by repressing the expression of BRCA1 and could be aimed at generation combination regimes to overcome PARPi resistance and enhance PARPi efficacy. Due to the poor pharmacokinetic profile and short half-life, the first-in-class BETi JQ1 was loaded into newly developed nanocarrier formulations to improve the effectivity of olaparib for the treatment of BRCAness cancers. First, polylactide polymeric nanoparticles were generated by double emulsion. Moreover, liposomes were prepared by ethanol injection and evaporation solvent method. JQ1-loaded drug delivery systems display optimal hydrodynamic radii between 60 and 120 nm, with a very low polydispersity index (PdI), and encapsulation efficiencies of 92 and 16% for lipid- and polymeric-based formulations, respectively. Formulations show high stability and sustained release. We confirmed that all assayed JQ1 formulations improved antiproliferative activity compared to the free JQ1 in models of ovarian and breast cancers. In addition, synergistic interaction between JQ1 and JQ1-loaded nanocarriers and olaparib evidenced the ability of encapsulated JQ1 to enhance antitumoral activity of PARPis.

## 1. Introduction

Poly (adenosine diphosphate ribose) polymerase inhibitors (PARPis) are targeted drugs used for the treatment of homologous recombination repair (HR)-defective tumours that have demonstrated antitumoral activity in several cancers harbouring germline and somatic BRCA1/2 mutations [1]. Cancer cells with BRCA1/2 protein deficiency or lacking functional mutations are more sensitive to PARPis than normal cells [2,3]. Three PARPis, olaparib, rucaparib and niraparib, were approved by the United States food and drug administration (FDA) in 2017 as a maintenance therapy for BRCA-mutated advanced ovarian cancer, and talazoparib was approved in 2018 for the treatment of deleterious germline BRCA-mutated HER2-negative metastatic breast cancer [4]. Although PARPis have shown considerable efficacy, their widespread use is restricted by various factors, including the development of acquired resistance due to the presence of compensatory pathways and their limited long-term tolerability [5]. In this context, there is a need for clinical strategies combining PARPis with additional chemotherapies, immune modulators or targeted agents with the ultimate aim of overcoming PARPi resistance and to enhance PARPi activity [6,7].

Modulation of transcription factors using epigenetic agents has demonstrated the ability to augment the efficacy of several types of compounds, including those that act on DNA repair mechanisms [8]. The bromodomain and extra terminal (BET) protein BRD4 promotes gene transcription through RNA polymerase II [9]. Specific BET inhibitors (BETis) reversibly bind the BRDs of BET proteins and prevent their interaction with acetylated histones and transcription factors [10]. Previous studies have suggested that BETis specifically modulate the expression of specific oncogenes [11]. The mechanism of action of BETis, including the first-reported-in-class JQ1, involves cell cycle arrest at G1 and an elevation of p27 [12]. Some BETis are currently in clinical development, showing varying rates of activity in several indications, including acute myeloid leukaemia [13] or multiple myeloma [14], among others.

Therapeutic inhibition of oncogenic vulnerabilities or signalling pathways has been demonstrated to increase the activity of PARPis and therefore overcome PARPi resistance [15]. For instance, combinatorial PARP inhibition with double blockade of AR [7], immunotherapy [16] or MEK inhibitors [17] has been reported to augment the efficacy of this family of agents. In this regard, BETis, such as JQ1, I-BET762, and OTX015, have been reported to act synergistically with olaparib in HR-proficient cancer cells [18]. The BET bromodomain inhibitor JQ1 synergized with the PARP inhibitor olaparib in BRCA1/2 wild-type ovarian cancer both in vitro and in vivo and was correlated with the suppression of both TOPBP1 and WEE1 [19]. Despite the antitumour effects of JQ1, its poor pharmacokinetic profile, low oral bioavailability and its short half-life [20] may limit its clinical development. To overcome this hurdle and improve its safety and efficacy profile, JQ1 has been encapsulated in different nanocarriers to reduce the delivery of the compound in non-tumoral areas and improve its pharmacokinetic and biodistribution profile [21,22,23,24,25]. A dual drug loading delivery system containing JQ1 has been explored to enhance efficacy and reduce systematic toxicity in different indications [26,27,28,29,30]. Synergic effects of JQ1 and temozolomide for the treatment of glioblastoma were reported using transferrin-functionalized NPs [31]. Immunoliposomes were successfully developed for the codelivery of irinotecan and JQ1 to elicit antitumor immunity in colorectal cancer [32]. JQ1 in combination with cyclin-dependent kinase 7 (CDK7) inhibitor THZ1 loaded in polyester NPs was reported as an effective molecular therapeutic option for the treatment of pancreatic ductal adenocarcinoma (PDAC) [33]. Peptide nanoparticles encapsulating a fixed ratio of olaparib and JQ1 were recently developed to treat non-BRCA mutant pancreatic cancer based on several recent studies that showed that JQ1 can completely disrupt the HR pathway by repressing the expression of BRCA1, RAD51 and BRD4 [34].

Herein, we propose the use of JQ1 encapsulated in polymeric- and lipid-based nanoparticles to enhance sensitivity to olaparib for the treatment of ovarian and breast cancer. JQ1 was encapsulated into polymeric nanoparticles based on FDA-approved polylactide and free-cholesterol lipid nanoparticles. The nanoparticles were fully characterized, the release profiles were studied and their stability under storage conditions and in vitro toxicity in non-tumoral cells were evaluated. In vitro cytotoxicity studies were performed to verify whether JQ1-loaded nanoparticles in combination with olaparib is an effective molecular therapeutic option for the treatment of ovarian and breast cancers. In summary, this work raises the possibility that a combination of olaparib and JQ1-loaded nanoparticles could have utility for the treatment of breast and ovarian cancers.

## 2. Materials and Methods

### 2.1. Materials

Poly-D,L-lactide (22,000 kDa) (PLA) was synthesized using Schlenk techniques under an argon atmosphere [35]. Zinc catalyst was prepared according to procedures described in the literature [36]. L-α-phosphatidylcholine, caprylic/capric triglyceride and Pluronic^®^ F-127 were purchased from Sigma-Aldrich (Madrid, Spain). (+)-JQ-1 (high-performance liquid chromatography (HPLC) ≥ 99.9%) was purchased from MedChemexpress. Olaparib (HPLC > 99%) was purchased from ChemCruz (Dallas, TX, USA).

### 2.2. Liposomal Formulation

Free-cholesterol liposomes (JQ1-LP) were formulated by solvent injection and evaporation method [37]. Briefly, 1 mg of (+)-JQ-1, 25 mg of L-α-phosphatidylcholine and 67 mg of capric/caprylic triglyceride were dissolved into 4.5 mL of acetone and subsequently dropped into 10 mL of Pluronic^®^ F-127 1% *w*/*v* solution. The mixture was homogenized with an Ultra-Turrax disperser for 10 min at 14k rpm. Acetone was removed in a rotary evaporator for 30 min at 40 °C.

### 2.3. JQ1-LP Polymeric Nanoparticle Formulation

JQ1-loaded NPs (JQ1-NPs) were formulated by double-emulsion method [38]. Briefly, 1 mg of JQ1 or olaparib and 10 mg of PLA were dissolved in 4 mL of dichloromethane, and 1 mL of Milli-Q water was added. The mixture was gently shaken in a vortex and homogenized in a sonicator homogenizer (Hielscher UP200S ultrasonic processor) for 1 min. The pre-emulsion was poured into 10 mL of poly (vinyl alcohol) (PVA) 1% *w*/*v* solution. The mixture was gently shaken in a vortex and homogenized in a sonicator homogenizer for 5 min. The organic solvent was removed at room temperature over the course of 1 h, and nanoparticles were collected after centrifugation for 20 min at 15 k rpm and 4 °C.

### 2.4. Characterization of Formulations

PLA was characterized using proton nuclear magnetic resonance (NMR) and gel permeation chromatography (GPC). NMR spectra were recorded on a Varian Inova FT-400 spectrometer (Palo Alton, CA, USA). GPC spectra were analysed on a PL-GPC-220 instrument. Size, polydispersity index and Z potential of nanoparticles were analysed using the dispersion light scattering technique on a Zetasizer Nano ZS instrument (Malvern Instruments, Malvern, UK). Data were analysed using the multimodal number distribution software included with the instrument. Particle size and morphology were monitored by both SEM and TEM. Prior to observation, particles were diluted in PBS and left to air dry at room temperature on metallic (Al/Cu) microscope stubs or grids. For SEM observation, specimens were coated with Au-Pt using an SC7620-Quorum Technologies (Lewes, UK) sputter coater in order to avoid charging-up problems on the specimen surface and to achieve better image resolution. Samples were observed on a Jeol 6490LV electron microscope operating at 20 kV. TEM images at higher resolution were acquired with a Jeol JEM 2100 TEM microscope operating at 200 kV and equipped with an Oxford Link EDS detector. Low-dose conditions were used for the observation to avoid damage to the specimens due to beam irradiation. The as-obtained images were analysed using Digital Micrograph™ software from Gatan (Pleasanton, CA, USA).

### 2.5. Efficiency and Loading Efficiencies

Loading efficiency of liposomes was calculated by according to the difference of drug feeding and the non-encapsulated drug found in the supernatant after dialysis in phosphate-buffered saline medium.

Loading efficiency (LE) and encapsulation efficiency (EE) of polymeric nanoparticles were determined using the NP destruction method [39,40] and calculated by means of the following equations:LE % = (weight of encapsulated JQ1 (mg))/(weight of total (JQ1 encapsulated + scaffold weight) (mg)) × 100%
EE % = (weight of encapsulated JQ1 (mg))/(weight of JQ1 feeding (mg)) × 100%

### 2.6. Stability of NPs

The NPs were stored at 4 °C. The average size (nm) and polydispersity index (PdI) of collected liposomes and polymeric NPs were monitored over time by dynamic light scattering (DLS) measurements.

### 2.7. In Vitro Drug-Release Studies

Collected liposomes and NPs were sealed in a dialysis membrane (12–14 kDa) and suspended in 50 mL of phosphate-buffered saline (pH 7.4 or 6.5) at 37 °C with magnetic stirring. The experiment was conducted in 3 replicates, and 1 mL of release medium was taken out at different time intervals to monitor the concentration by UV-vis spectroscopy.

### 2.8. Cell Culture and Drug Compounds

Triple-negative breast cancer cell lines MDA-MB-231 and BT549 were grown in DMEM and RPMI medium, respectively. Ovarian cancer OVCAR8 and SKOV3 cell lines were grown in DMEM medium. Normal epithelial cell line HaCat was grown in DMEM medium. All medium contained a high glucose concentration (4.5 g/L) and L-glutamine (2 mmol/L). All media were supplemented with inactivated foetal bovine serum (10% FBS) and antibiotics (100 U/mL penicillin and 100 μg/mL streptomycin). Cells were grown on adherent culture and maintained at 37 °C in a humidified atmosphere in the presence of 5% CO_2_. All cell lines used in the present study were provided by Dr. J. Losada and Dr. A. Balmain (from the ATCC) in 2015. All cell culture media and supplements were obtained from Gibco (Thermo Fisher, Waltham, MA, USA).

### 2.9. MTT Proliferation Assay

The drug effect on cell proliferation was evaluated using thiazolyl blue tetrazolium bromide (MTT, Sigma Aldrich) colorimetric assay. Cells were seeded on 48-wells plates (5000 cells/well) and treated with indicated increase doses of each drug, either alone or in combination, JQ1-loaded nanodevices and empty nanocarriers. After 72 h of treatment, cell medium was replaced with MTT solution (red phenol-free DMEM with MTT, 0.5 mg/mL) for 1 h at 37 °C. DMSO was then used to solubilize the formazan precipitates. Absorbances at 555 nm values were recorded in a spectrophotometer multiwell plate reader (BMG labtech, Ortenberg, Germany). A wavelength of 690 nm was used as a control reference.

### 2.10. EC_50_ Values

Half maximal effective concentration (EC_50_) was calculated using the Sigmoidal, 4PL, x is log (concentration) equation in the dose–response curve of each drug. GraphPad Prism 7.0. software was used.

### 2.11. Cell Cycle and Apoptosis Assay

For cell cycle analyses, cell lines were plated at 100,000 cells per well in 6 multiwell plates. After 24 h, wells were treated with an EC_50_ dose of JQ1 and JQ1-LP for 24 h. To fix the cells, the cells were incubated for 30 min in 70% ethanol. Then, pellets were washed in 2% BSA in PBS and stained with Propidium iodide/RNAse staining solution for 1 h at 4 °C under dark conditions (Immunostep S.L., Salamanca, Spain).

For apoptosis analysis, the same number of cells was treated for 72 h. Then, adherent and floating cells were washed with PBS and incubated in annexin V binding buffer (Immunostep S.L.) for 1 h in the dark with annexin V and PI staining solution (Immunostep S.L). The percentage of dead cells was determined considering early apoptotic (annexin V-positive, PI-negative), late apoptotic (annexin V-positive and PI-positive) and residual necrotic (annexin V-negative, PI-positive) cells, which were included as dead cells in the analysis.

Flow cytometry assays were evaluated in a FACSCanto II flow cytometer (BD Bio-sciences). Histogram and dot-plot representation was performed using Flowing software version 2.5.1. (Perttu Terho, Turku Center for Biotechnology, University of Turku, Finland).

### 2.12. Synergy Studies

To analyse whether the effect of the compounds was additive, synergic or antagonist, varying doses of JQ1 or JQ1-encapsulated and olaparib were combined, and Calcusyn 2.0 software (Biosoft) was used. The Chou–Talalay algorithm was applied to obtain the combination index (CI), indicating synergistic effect (<1), additive effect (=1) and antagonistic effect (>1).

### 2.13. Statistical Analyses

Statistical analysis was performed using GraphPad Prism software (version 7.0) (GraphPad Software Inc., La Jolla, CA, USA). Data are presented as the mean ± standard deviation of at least three independent experiments. In order to identify statistically significant differences between treatments, the unpaired t-test for independent samples was used, considering a level of significance of 95%, using the following legend: * *p* ≤ 0.05; ** *p* ≤ 0.01 and *** *p* ≤ 0.001.

## 3. Results

### 3.1. Formulation, Characterization, Releases Profiles and Stability of Nanodevices

JQ1-loaded, cholesterol-free, lipid-based nanoparticles (JQ1-LP) were generated via an emulsification method. A JQ1:lipid ratio of 1:92 *w*/*w* and a temperature of 40 °C were used for JQ1 encapsulation (Figure 1A). The building blocks selected for the generation of JQ1-loaded polymeric NPs were the FDA polymers polylactide (PLA), and a double emulsion method was adapted and optimized for the loading of JQ1. The formulation was optimized until the average size and PdI were suitable for further studies. NPs were characterized by dynamic light scattering (DLS) technique, scanning electron microscopy (SEM) and transmission electron microscopy (TEM). Table 1 shows the average size, polydispersity (PdI) and Z potential of non-loaded and JQ1-loaded nanodevices. DLS studies revealed a hydrodynamic radius (R_H_) for formulations close to 60 nm for lipid-based nanodevices and close to 120 for polymeric-based nanoparticles. The loading of JQ1 into the nanoplatforms did not alter the physical parameters of the formulations.

Both SEM and TEM images revealed the presence of spherical morphologies and confirmed the results obtained via DLS, i.e., LP-JQ1 particles were smaller than JQ1-NPs (Figure 1B,C). In some cases, larger particles were also detected due to merging processes upon solvent evaporation. The larger particles observed exhibited deviation from sphericity, as highlighted in Figure 1B. The encapsulation efficiency (EE%) and loading efficiency (LE%) of 98% were determined by analysis of the supernatant after 30 min of dialysis in PBS at room temperature for JQ1-LP. The LE% for JQ1-NPs was 2.6% for the optimized formulation, with an encapsulation efficiency of 45%.

Physical stability of JQ1-LP and JQ1-NPs in PBS at 4 °C was monitored by DLS measurements (Figure 2A). No significant increments on particle size and polydispersity denoted satisfactory stability against aggregation for both formulations. In vitro release studies were conducted for each formulation in pH 7.4 at 37 °C. JQ1-LP showed a strong burst release in the beginning of the experiment, followed by a slow release, which did not exceed 60% after 50 h. This might be secondary to a lack of stiffness due to a cholesterol-free formulation. In contrast, a sustained release in which complete JQ1 delivery was achieved within 700 h was observed for JQ1-NPs.

pH variations are present in the tumour microenvironment. Cancerous tissue is slightly acidic. This pH gradient has been widely used to design nanosystems and deliver therapeutics to tumour tissues [41]. To gain insight into the release of JQ1 from nanodevices in such an acidic microenvironment, JQ1 release at pH 6.5 PBS from both formulations was also assessed (Figure 2B). Only a slightly quick release of JQ1 was observed for lipid-based nanoparticles at pH 6.5, and no significant changes were observed for polymeric nanoparticles.

### 3.2. Antitumour Effects of JQ1 and JQ1-Loaded Nanocarriers on Cell Proliferation

To study the ability of both JQ1-loaded formulations to inhibit cell proliferation, four cell line models representative of triple-negative breast and ovarian cancers were evaluated: MDA-MB231 and BT549 and OVCAR8 and SKOV3, respectively. The induction of cell toxicity in dose–response cell survival assays was first evaluated. As shown in Figure 3A, after 72 h treatments, the liposomal formulation showed equal activity compared with the free drug in the TNBC subtype, with EC_50_ showing no significant differences between treatments (Figure 3B). In the ovarian cancer cell line models, the liposomal formulation demonstrated a higher antitumoral activity at different doses (Figure 3C), with a lower EC_50_ for JQ1-LP compared with the free compound, although these differences did not reach a statistically significant level (Figure 3D). JQ1-LP display equal signs of antiproliferative activity compared with free JQ1, reaching EC_50_ values of approximately 500 nM for JQ1 and JQ1-LP in MDA-MB-231 and 1500 nM in BT549.

Next, the antiproliferative effect of JQ1-NPs was evaluated in the same cell line models. JQ1-NPs showed a higher antiproliferative effect in TNBC cell lines, reaching EC_50_ values of approximately 100 nM in MDA-MB-231 and 300 nM in BT549 (Figure 3A), with lower EC_50_ values compared with the free compound (Figure 3B). In the two cell line models of ovarian cancer, OVCAR8 and SKOV3, the differences reached a statistically significant level, with EC_50_ values much lower than those for JQ1 (Figure 3C,D). These data indicate that JQ1-NPs are more potent than free JQ1.

### 3.3. Effect on Proliferation of Non-Loaded Nanocarriers

To study the contribution of non-loaded nanocarriers, including empty liposomes and empty polymeric NPs, to the previously observed antiproliferative effects, the four tumoral cell lines, BT549, MDA-MB231, OVCAR8 and SKOV3, were treated with increasing concentrations of both formulations. As shown in Figure 4A, empty NPs displayed no effect on proliferation, and empty liposomes exhibited only a slight antiproliferative effect, which was more profound in BT549. Equivalent results were observed when these compounds were studied in the non-transformed cell line HaCat (Figure 4B). Next, the antiproliferative activity of loaded formulations and the free compound in the non-transformed cell line, HaCat, was explored. Doses of JQ1-LP, JQ1-NPs and the free JQ1 as high as 400 nM did not reach the EC_50_ (Figure 4C).

### 3.4. Synergistic Interaction between JQ1, JQ1-LP or JQ1-NPs and Olaparib in Representative Models of Triple-Negative Breast and Ovarian Cancers

Given the synergistic interaction observed between JQ1 and olaparib as free compounds in some preclinical models [19], we evaluated whether the formulation of JQ1 as liposomes or nanoparticles would maintain the same level of activity. As shown in Figure 5 and Figure 6, the combination of both agents was synergistic, independent of the formulations used. Combination indices (CIs) were below 1, except for JQ1 alone in BT549 and OVCAR8 and JQ1-LP in BT549. However, the combination of JQ1-loaded nanocarriers with olaparib showed an enhanced effect at different doses, particularly in both evaluated TNBC cell lines, MDA-MB-231 and BT549 (Figure 5B). Similar findings were observed in the ovarian cancer cell lines OVCAR8 and SKOV3, showing that both formulations with JQ1, when combined with olaparib, exhibited considerable synergistic action (Figure 6B). These results demonstrate that the combination of olaparib with the encapsulated formulations displayed an enhanced synergistic interaction relative to the combination of single compounds.

### 3.5. Free JQ1 and Encapsulated Formulations of JQ1 Induce G0/G1 Arrest

The inhibition of BET has previously shown an antiproliferative effect in TNBC and ovarian cancers [12]. The mechanism of action of JQ1 and the encapsulated forms of JQ1 were studied in MDA-MB-231 and OVCAR8 as human cell models for TNBC and ovarian cancer, respectively (Figure 7). In the breast cancer cell line model, MDA-MB-231, only JQ1-NPs at the highest concentrations were able to induce cell cycle arrest at G0/G1 (Figure 7A). On the other hand, in the ovarian model, OVCAR8, cell cycle arrest was observed with both formulations, i.e., JQ1-LP and JQ1-NPs (Figure 7B). In both cell line models, JQ1 in its free form did not induce cell cycle arrest at the two doses used, i.e., 200 nM and 400 nM (Figure 7A,B). Previous experiments performed by our group demonstrated that the arrest in G1 with JQ1 was observed with doses higher than 500 nM and after a minimum of 24 h [12,42]. Globally, these data suggest that JQ1 formulated as liposomes or nanoparticles can have a similar or slightly augmented effect on G1 cell cycle arrest as the free forms of JQ1.

### 3.6. Cell Death Is Highly Induced with the Encapsulated Formulations of JQ1 in Combination with Olaparib

Next, we explored how the combination of free JQ1 or as an encapsulated form (liposomes or nanoparticles), when combined with olaparib, was able to induce cell death in cellular models. Our previous data suggested that JQ1, when combined with chemotherapies, was able to rapidly augment cell death, although its effect as a single agent was modest [12]. To evaluate this effect, we treated cell lines with JQ1-LP, JQ1-NPs and the free form in combination with olaparib in MDA-MB-231 and OVCAR8. At the evaluated concentrations, the combination of free JQ1/JQ1-LP/JQ1-NPs and olaparib resulted in an increased in apoptotic cell death (annexin V-positive cells), and the encapsulated forms demonstrated equal activity as free formulations in both cell lines (Figure 8A,B and Appendix A). These data show that the lipid- and polymeric-based formulations maintain the same potency to induce cell death as the free compounds.

## 4. Discussion

Targeting oncogenic vulnerabilities, such as DNA repair mechanisms, in tumours with special susceptibility is a main objective of cancer research. This is the case for the synthetic lethality approach using PARPis in tumours with mutations in the BRCA1 or BRCA2 genes [4]. This strategy has proven to be efficient in the clinical setting, but limitations have also been observed, including the presence of primary or secondary resistance, as well as high dosage requirements limiting the efficacy for the treatment of some patients [43]. In this context, combinations with other agents have shown potential, such as those acting on epigenetic pathways [44]. A specific family of agents that acts on epigenetics are those acting on BRD proteins, such as BETis.

The first-in-class BETi, JQ1, has shown efficacy in preclinical studies and clinical trials [45]. The anticancer potential of JQ1 is based on the blocking of BRD4, which is involved in the proliferation of numerous malignant tissues. However, its translation to clinical practice is constrained by its short half-life, poor pharmacokinetic profile, and low oral bioavailability. Therefore, the finding of a right nanocarrier may improve its potential for clinical development. The first part of our work entails the encapsulation of JQ1 in nanocarriers. The extended release of JQ1 from nanodevices could result in augmentation of its therapeutic index and improvement of several PK properties. This was evidenced when JQ1 was encapsulated into polymeric NPs to improve the anticancer efficacy against preclinical models of TNBC in vitro and in vivo [21]. A non-FDA-approved biodegradable poly(disulphide amide) was successfully designed for the generation of polymeric NPs for targeted delivery of JQ1 for gallbladder cancer treatment [23].

Liposomes are considered efficient vehicles for drug administration, as they are biocompatible and biodegradable, and their membrane structure facilitates cellular uptake [46] and can encapsulate hydrophobic drugs, such as JQ1. In this regard, cubosomes, the most advanced generation of lipid-based nanoparticles, have attracted considerable attention due to their internal nanostructures, which offer structural versatility to improve stability and encapsulation efficiency [47,48]. On the other hand, polymeric nanostructures are nanoparticles with potential for a prompt translation to clinical practice [38]. Bearing in mind the potential translation of JQ1-loaded nanocarriers, polymeric nanoparticles based on the FDA-approved PLA, as well as cholesterol-free liposomes, were chosen for our studies. PLA was chosen as a polymeric raw material for the generation of NPs to provide flexibility in terms of physicochemical parameters, cargo and release of JQ1 at nanoscale size, as well as potential for the selective addressing and controlled delivery of JQ1 [38]. PLA NPs are biodegradable, biocompatible, highly stable during storage and FDA-approved. Therefore, JQ1-loaded polymeric- and lipid-based nanocarriers were generated following procedures previously reported by our research group [37,39,40]. The liposomes and polymeric NPs obtained were in the R_H_ range of 60 to 120 nm, with very low PDI values, which were similar to values reported in previously published works involving the entrapment of JQ1 in polymeric NPs [21,23,27]. Both nanocarriers showed very high stability over 7-day storage. JQ1-LP showed an exponential release profile, with an initial burst release of no more than 60%. JQ1-NPs sustained a drug release profile over time, achieving release after 50 h, in contrast with JQ1-LP, the release of which was slowed down after 8 h. No significant differences were observed when release experiments were conducted under slightly less acidic conditions. Once the JQ1-loaded nanocarriers were fully characterized by measurement of size, polydispersity index and Z potential, as well as morphology reported by SEM and TEM and stability of the nanocarriers and release profile of the studied JQ1, in the second part of the work, we assayed the ability of our JQ1-loaded nanocarriers to target different models of cancer cells: MDA-MB231 and BT549 as models for TNBC and two ovarian cancer cell lines, OVCAR8 and SKOV3. In vitro assays confirmed that JQ1-loaded nanocarriers retain the ability of free JQ1 to inhibit proliferation at different concentrations. Empty nanocarriers did not show any toxicity in the tumour cells and in non-transformed cells at any concentration in any of the assays used. As a proof of their safety, JQ1-loaded nanocarriers did not show a significant toxicity in non-transformed cells, showing even less toxicity than the free drug. Herein, we confirm that the encapsulation of JQ1 improves drug efficacy. Loading JQ1 onto nanoparticles might extend its biodistribution by delaying its release and reducing its effect on non-transformed tissues.

It was previously reported that JQ1 synergized with olaparib in BRCA1/2 wild-type ovarian cancer both in vitro and in vivo and correlated with the suppression of TOPBP1 and WEE1 [19]. Here, we assessed the synergic effect of free JQ1 in comparison to encapsulated JQ1 and olaparib to target representative models of human BRCAness tumours. JQ1-NPs and JQ1-LPs showed a synergistic interaction with olaparib in both evaluated TNBC and ovarian cell lines. These results confirm that the encapsulated form of JQ1 enhances antitumoral activity when combined with PARPis for the treatment of triple-negative breast and ovarian cancers. How to overcome resistance to chemotherapy, particularly in indications such as TNBC, is a major challenge. Recently, some biological mechanisms have been described, such as the presence of liver X receptor alpha [49] or SEMA6D/miRNA 195 in triple-negative tumours [50]. In this context, the proposed combination described in this article could target those mechanisms, improving the efficacy of current therapeutic strategies.

An interesting finding is the fact that combination of JQ1-encapsulated NPs and olaparib induced a profound antitumoral effect in TNBC and ovarian cancer cell lines and that the mechanism of action was mediated by apoptotic cell death. JQ1 formulated as liposomes or nanoparticles can have a similar or slightly augmented effect on G1 cell cycle arrest as the free forms of JQ1. This mechanism of action is in line with that described with a free JQ1 formulation [12].

The ability of JQ1-encapsulated NPs to enhance the efficacy of other drugs is in line with results reported in previous studies based on the codelivery of JQ1 and other therapeutic agents [27,28,31,32,33,34]. JQ1 and temozolomide codelivery increased DNA damage and apoptosis in gliomas [31]. JQ1 and shikonin can target the multiple components of the tumour immune microenvironment [28]. Compared to free drugs, the codelivery of JQ1 and the cyclin-dependent kinase 7 inhibitor THZ1 significantly enhanced tumour-inhibition effects in a gemcitabine-resistant pancreatic ductal adenocarcinoma patient-derived xenograft model [33]. To the best of our knowledge, the synergic effect of encapsulated JQ1 and free PARPis for the treatment of TNBC and ovarian cancer has not been reported.

## 5. Conclusions

Herein, we described novel formulations of the BETi JQ1, which, when combined with olaparib, augmented antitumoral efficacy against triple-negative breast and ovarian cancers, maintaining the same mechanism of action. This work paves the way for future studies combining JQ1 nanoparticles with agents acting on specific tumour vulnerabilities and demonstrates that nanoformulations of targeted agents can be used in combination to augment antitumoral activity. Overall, these novel formulations may represent an efficient and safe JQ1 delivery alternative to enhance the efficacy of olaparib for the treatment of BRCAness tumours.

## Figures and Tables

**Figure 1 cancers-14-04474-f001:**
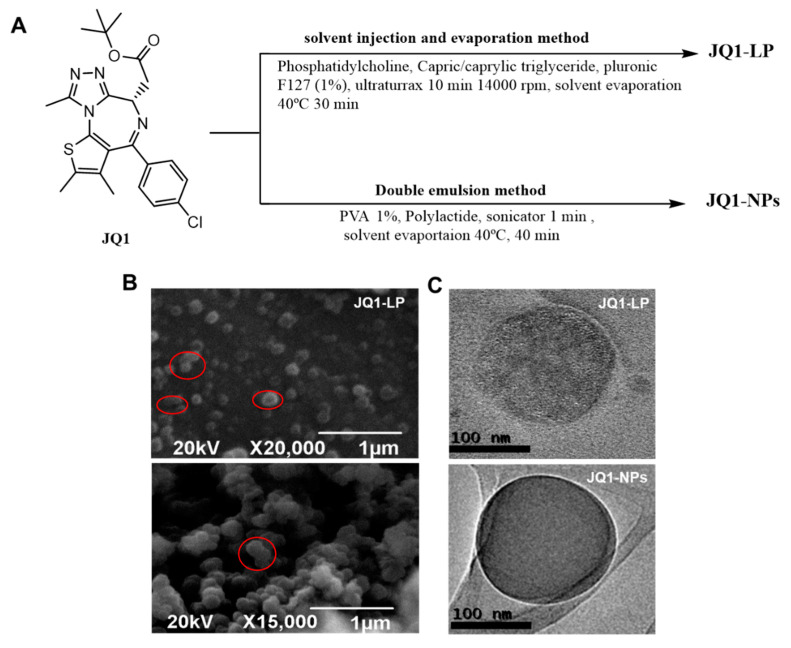
Formulation and morphology of JQ1-LP and JQ1-NPs. (**A**) Schematic formulation of nanodevices. (**B**) SEM images of nanodevices (scale bar = 1 µm). (**C**) TEM images of nanodevices (scale bar = 100 nm).

**Figure 2 cancers-14-04474-f002:**
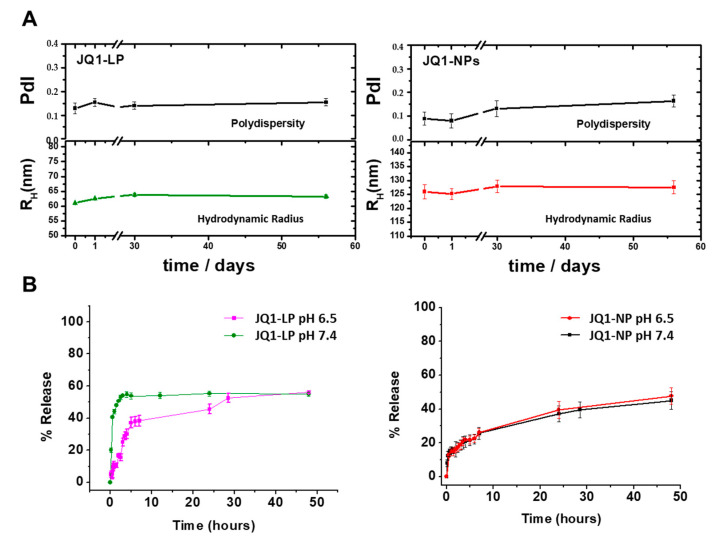
Storage stability and in vitro release profiles of JQ1-loaded nanodevices. (**A**) DLS analysis showing the stability of nanodevices. (**B**) Release kinetics of nanodevices in PBS (pH 7.4 and pH 6.5) at 37 °C. Data are expressed as mean ± SEM from at least three independent experiments.

**Figure 3 cancers-14-04474-f003:**
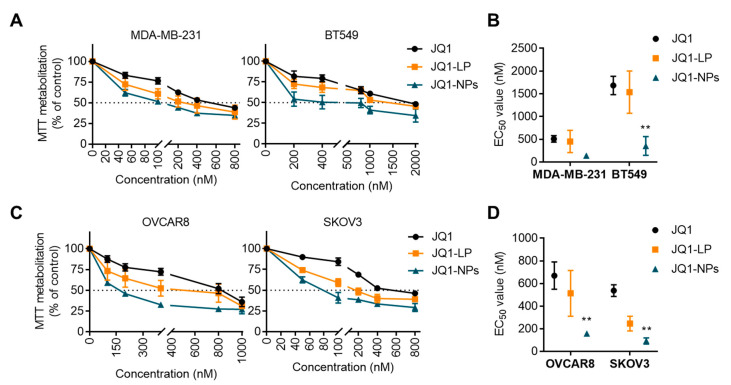
The effect of JQ1 nanocarriers in TNBC and ovarian cancer cell lines. Dose–response curves comparing JQ1 free, JQ1-LP and JQ1-NPs in MDA-MB-231 and BT549 (TNBC) cell lines (**A**) and OVCAR8 and SKOV3 (ovarian cancer) cell lines (**C**). Cells were seeded in p48 plates (5000 cells/well) and treated for 72 h with indicated doses of compound. EC_50_ values calculated by interpolation using GraphPad Prism 7.0 software shown in (**B**) (TNBC) and (**D**) (ovarian cancer). Statistical analysis between control JQ1 versus JQ1-LP or JQ1-NPs are shown. ** *p* ≤ 0.01.

**Figure 4 cancers-14-04474-f004:**
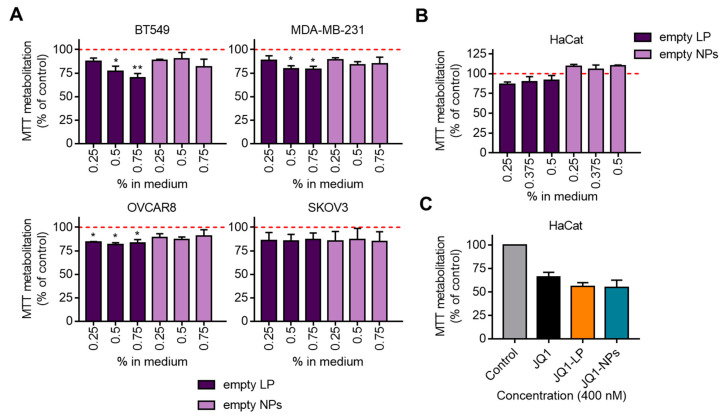
Evaluation of toxicity of empty encapsulations and their effect in normal-keratinocyte cells. (**A**) Cells treated with non-loaded formulations with similar medium percentage (%) relative to the previous proliferation assay (with JQ1 encapsulation) (0.25% in medium = 250 nM; 0.5% in medium = 500 nM; 0.75% in medium = 750 nM). (**B**) Human keratinocyte cell line HaCat treated with empty liposomes and empty nanoparticles (0.25% in medium = 250 nM; 0.375% in medium = 375 nM; 0.5% in medium = 500 nM). (**C**) HaCat cells were treated with JQ1-free, JQ1-LP and JQ1-NPs at 400 nM. * *p* ≤ 0.05 and ** *p* ≤ 0.01.

**Figure 5 cancers-14-04474-f005:**
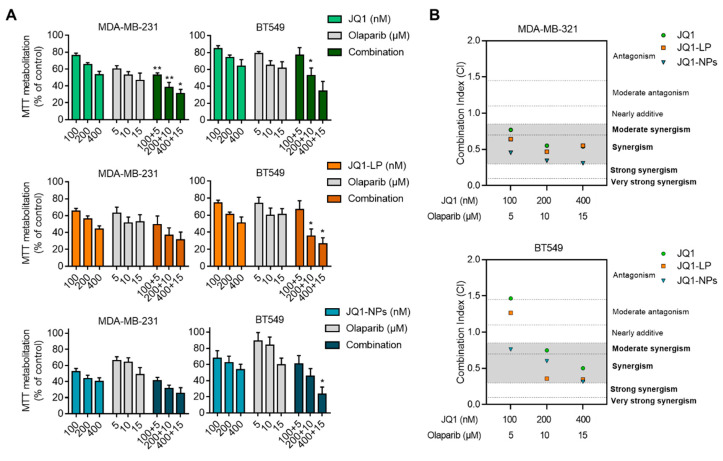
Evaluation of JQ1 encapsulation and olaparib combination in breast cancer cell lines. (**A**) MDA-MB-231 and BT549 cell lines treated with JQ1-free (upper), JQ1-LP (middle) and JQ1-NPs (lower) alone or in combination with olaparibfor 72 h, followed by MTT assays. (**B**) Synergistic studies to evaluate the effect of JQ1-free NPs or encapsulated with olaparib. Combination index (CI) for the different drug combinations obtained using CalcuSyn from viability values obtained in an MTT assay (**A**). CI values lower than 0.8 indicate synergistic action. * *p* ≤ 0.05 and ** *p* ≤ 0.01.

**Figure 6 cancers-14-04474-f006:**
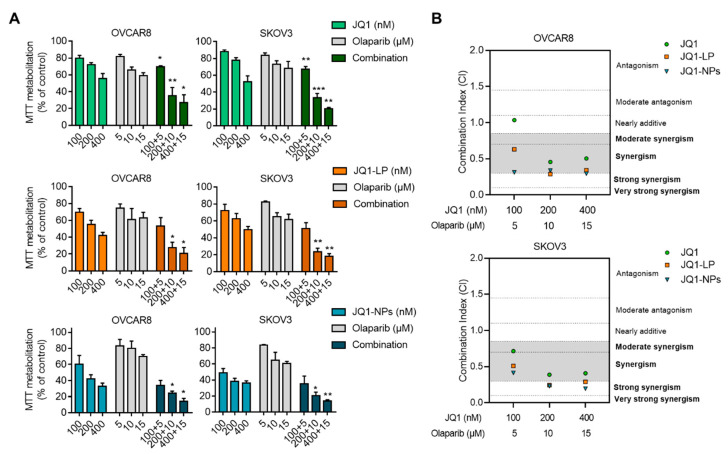
Evaluation of JQ1 encapsulation and olaparib combination in ovarian cell lines. (**A**) OVCAR8 and SKOV3 cell lines treated with JQ1-free (upper), JQ1-LP (middle) and JQ1-NPs (lower) alone or in combination with olaparib-free NPs for 72 h, followed by MTT assays. (**B**) Synergetic studies to evaluate the effect of JQ1-free NPs or encapsulated with olaparib-free NPs. Combination index (CI) for the different drug combinations obtained using CalcuSyn from viability values obtained in an MTT assay (**A**). CI values lower than 0.8 indicate synergistic action. * *p* ≤ 0.05; ** *p* ≤ 0.01 and *** *p* ≤ 0.001.

**Figure 7 cancers-14-04474-f007:**
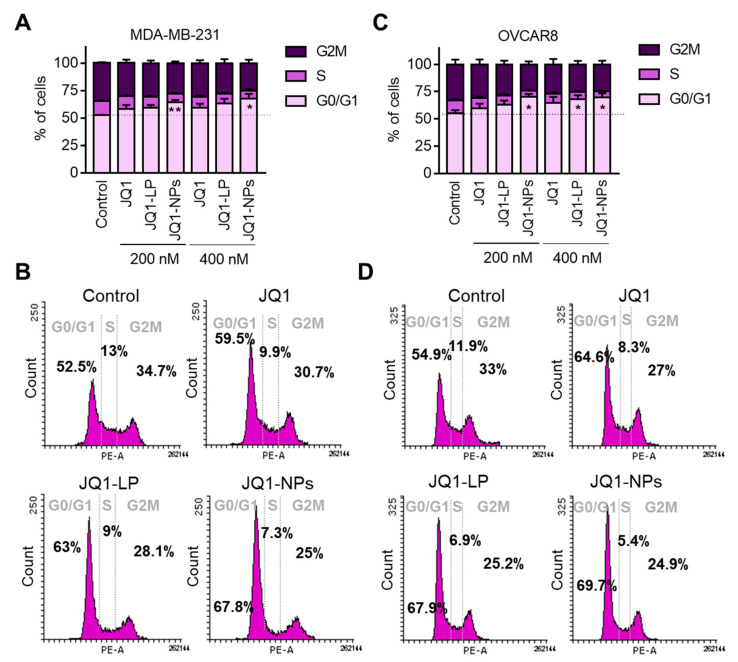
Cell cycle analyses show free JQ1 and JQ1 encapsulation increased G0/G1 phase arrest. MDA-MB-231 breast cancer cells (**A**) and OVCAR8 ovarian cancer cells (**C**) were treated for 24 h with 200 or 400 nM of JQ1 (free), JQ1 liposomes and JQ1 nanoparticles. Later, cells were fixed and stained with PI for cell cycle evaluation. (**B**,**D**) Histogram representing the percentage of cells in each cell cycle phase after each treatment (400 nM). * *p* ≤ 0.05, ** *p* ≤ 0.01.

**Figure 8 cancers-14-04474-f008:**
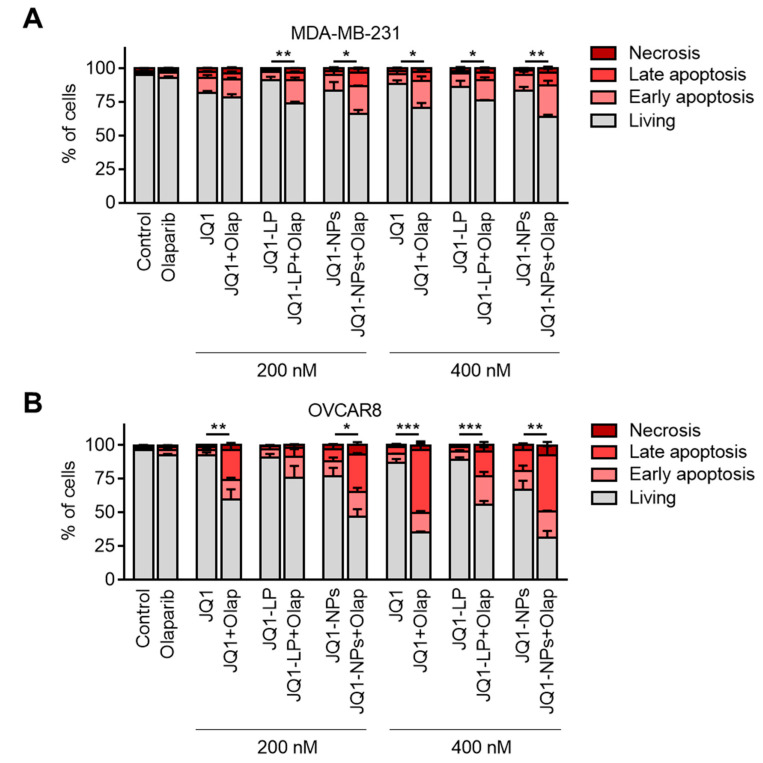
JQ1 free and JQ1 encapsulated forms synergize with olaparib. MDA-MB-231 breast cancer cells (**A**) and OVCAR8ovarian cancer cells (**B**) treated for 72 h with 200 or 400 nM of JQ1 (free), JQ1-LP and JQ1-NPs alone or in combination with olaparib (10 µM). * *p* ≤ 0.05; ** *p* ≤ 0.01 and *** *p* ≤ 0.001.

**Table 1 cancers-14-04474-t001:** Characterization of loaded and non-loaded nanodevices.

Formulation	R_H_ (nm)	PDI	Z Potential (mV)
JQ1-LP	61.04 ± 0.14	0.13 ± 0.02	−34.20 ± 1.75
JQ1-NPs	125.19 ± 1.67	0.08 ± 0.01	−15.93 ± 0.60
NPs	138.63 ± 9.29	0.09 ± 0.02	0.94 ± 0.27
LIP	65.21 ± 1.08	0.08 ± 0.02	−35.00 ± 2.27

Hydrodynamic radius (R_H_), polydispersity index (PdI), JQ1-loaded liposomes (JQ1-LP), JQ1-loaded polymeric nanoparticles (JQ1-NPs). Errors are 2σ.

## Data Availability

The data presented in this study are available in this article.

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
