# Peer review of "Enhanced Antitumoral Activity of Encapsulated BET Inhibitors When Combined with PARP Inhibitors for the Treatment of Triple-Negative Breast and Ovarian Cancers"

_cancers, 2022, doi:10.3390/cancers14184474_

Round 1
Reviewer 1 Report
The manuscript submitted to cancers by Juan A. et al. explores the combination of the PARP inhibitor Olaparib and the BET inhibitor JQ1 in two triple negative breast cancer cell lines and two ovarian cell lines. To overcome JQ1 pharmacokinetic issues they encapsulated the drug into liposomes or polylactide polymeric nanoparticles. Results are restricted to in vitro analyses and include characterisation of the nanodevices, cell proliferation and drug combination assays, and cell cycle and apoptosis studies. JQ1 has been previously encapsulated in other types of nanocarriers, the novelty here is that the authors have used free-cholesterol liposomes and FDA-approved polymeric nanoparticles to improve translation of their results into the clinic. While JQ1 is not currently in clinical use or undergoing clinical trials, combination therapies using BET inhibitors are of great interest in oncology and the results reported in this manuscript are relevant to the field.
Major issues:
- Figures 7 and 8 show cell cycle and PI/AnnexinV flow cytometry analyses but there is no description of flow cytometry in the “Materials and Methods” section.
- In the experiments with non-loaded nanocarriers the authors need to indicate the equivalence between the amount of empty nanocarrier (% in the medium) and the amount of drug-loaded nanocarrier (concentration nM). Otherwise, there is no way of knowing whether the concentrations of empty nanocarrier tested are within the range of those used for the rest of the studies.
- Line 289: The authors must revise their conclusion “… JQ1-loaded nanocarriers did not show significant toxicity in non-transformed cells”. Figure 4C shows that JQ1-LP 400 nM induces nearly 50% cell death of non-tumour keratinocytes HaCat. EC50 values for JQ1-LP in the four tumour cell lines tested are within that range (or even higher).
- Lines 407-408, Discussion: The authors must revise their statement “We showed that both nanocarriers improved the ability of free JQ1 to inhibit proliferation at different concentrations”. Encapsulation of JQ1 into liposomes did not significantly increase cell proliferation inhibition compared to equivalent doses of free drug (Figure 3).
- Figures 5 and 6: It is not indicated whether asterisks (p values) in the bars of the Combination (free JQ1/JQ1-LP/JQ1-NPs + Olaparib) are compared to free JQ1/JQ1-LP/JQ1-NPs alone or to Olaparib alone. Nevertheless, it is not coherent that if the MTT data show no (significant) differences between the drug Combination and each drug alone (free or encapsulated) the combination index (calculated using those same data) indicates synergism. Authors need to explain this point or to revise their data and their conclusions in the Discussion (lines 419-421: “JQ1-NPs and JQ1-LP showed a remarkable synergistic interaction with olaparib when 419 comparing with free JQ1, particularly in both TNBC cell lines evaluated, MDA-MB-231 420 and BT549”).
- Figure 7 B and D: analyses of cell cycle data should not be done using gating. The authors should use cell cycle analysis software (FCS Express, ModFit…).
Minor issues:
- The quality of the graphics in Figures 3, 4, 5, 6 and 8 must be improved. It is a challenge to read the text in some of the figures.
- Lines 44-45, Abstract: “In addition, synergistic interaction between JQ1 and JQ1-loaded nanocarriers and Olaparib evidenced the ability of encapsulated JQ1 to enhance antitumoral activity of BETis”. Should it read PARPis?
- Line 112: it seems that two references are missing.
- Line 244: reference to Figure 2C: there is no Figure 2C.
- Lines 276-295, section 3.3. Effect on proliferation of non-loaded nanocarriers and Figure 4: The authors are suggested to change the naming “LP-free” and NPs-free” to “empty LP” and “empty NPs” to avoid confusion with “JQ1 free/free JQ1”, which they use to refer to the non-vehiculised form of JQ1.
- Lines 365-367: “…synthetic lethality approach using PARPis in tumors with mutations at the BRCA1 or BRCA2 genes [40].” Reference (self-citation) does not correspond.
- Line 417: “…correlated with the suppression of TOPBP1 416 and WEE1 [16].” Reference does not correspond.
- Line 567: I cannot find reference 39 (self-citation) anywhere in the text.
- Figure 2: panels A and B are not indicated.
- Figure 4 A and B: colours for LP-free and NPs-free are too similar and cannot be distinguished.
- Figure 8: has four panels A-B-C-D, but the legend only describes A and B.
- The manuscript needs a thorough revision of English language and style. There are frequent errors in the use of was/were, has/have or grown/growth, among others. “Harboring” is repeated twice in line 52, “NPs is much potent” …
- The authors should use either “Olaparib” or “olaparib”.
- Section 3.6 and Figure 8: The authors might want to comment in this section that, at the concentrations tested, the combination of free JQ1/JQ1-LP/JQ1-NPs + Olaparib increases apoptotic cell death (Annexin V positive cells). There is a mention to this later in the Discussion.
Author Response
Attached PDF with the reply

Reviewer 2 Report
The authors have presented an impressive study of BET inhibitors as a therapeutic approach against ovarian cancers and TNBC. However, there are some minor comments I would like to suggest in order to improve the articles before publishing.
1. The abbreviation of JQ1-LP, JQ1-NPs etc should be included in the table 1 captions for ease of readers.
2. The color combinations used in panel A and B for figure 4 should be changed. As it could be misleading if printed in black and white. Also in online version the two colors are quite similar and close.
3. In figure 2, the legend spelling of hydrodinamic radius should be changed to ‘hydrodynamic radius’.
4. The combination index (CI) graph in fig5 and fig6 is well appreciated and summarizes the effect of the combination therapy quite well.
5. Similar to the lipidic nanocarriers, cubosomes which are the 2nd generation of lipidic nanocarriers are currently gaining attention for targeted delivery of therapeutics in cancers. The authors is encouraged to discuss this with reference to articles such as (https://doi.org/10.1021/acsami.1c21655) and (https://doi.org/10.1021/acs.molpharmaceut.2c00439).
6. Has the authors studied the release of the JQ1 from the LP and NPs at pH around 6.5? If would be interesting if the authors throw some light on it as cancers or tumor microenvironment has slightly acidic pH compared to normal tissue. And if there is any change of the release kinetics at different pH?
7. The therapeutic possibility of using BETis to treat the drug resistance triple negative breast cancer (TNBC) resistant could be a promising approach. The authors should include this possibility in discussion for treating drug resistance TNBC which are caused due to some newly identified novel genes such as LXRalpha and SEMA6D/miRNA 195.
8. Fig 8 is low resolution. As the authors are showing the bar graphs which summarizes the dot plot of Annexin V FACS assay. So, the bar graphs could stay in the main figure whereas the dot plots could be moved to Supplementary figures. That would improve the resolution of the overall figure.
Author Response
Attached PDF with the reply

Round 2
Reviewer 1 Report
Reference line 406 should be [19]?
The manuscript would benefit from some English language editting.
Author Response
Comments and suggestions for Authors
Reference line 406 should be [19]?
Authors response. We regret and apologize for the mistake. The reference has been replaced by Karakashev, S.; Zhu, H.; Yokoyama, Y.; Zhao, B.; Fatkhutdinov, N.; Kossenkov, A.V.; Wilson, A.J.; Simpkins, F.; Speicher, D.; Khabele, D.; et al. BET Bromodomain Inhibition Synergizes with PARP Inhibitor in Epithelial Ovarian Cancer. Cell Reports 2017, 21, 3398–3405, doi:10.1016/j.celrep.2017.11.095 (Reference 19).
The manuscript would benefit from some English language editing
Authors response. The revised manuscript was proofread again before resubmitting.